

# Particulate matter collection by honey bees (*Apis mellifera*, L.) near to a cement factory in Italy

Marco Pellecchia[1] and Ilaria Negri[2]

[1] Koiné—Environmental Consulting S.n.c., Montechiarugolo (Parma), Italy
[2] Department of Sustainable Crop Production, Università Cattolica del Sacro Cuore, Piacenza, Italy

## ABSTRACT

Industrial activities play a key role in the economic well-being of a country but they usually involve processes with a more or less profound environmental impact, including emission of pollutants. Among them, much attention has been given to airborne particulate matter (PM) whose exposure is ubiquitous and linked with several adverse health effects mainly due to its size and chemical composition. Therefore, there is a strong need to exploit monitoring systems for airborne PM able to provide accurate information on the potential health hazards and the specific emission sources for the implementation of adequate control strategies. The honey bee (*Apis mellifera*, L.) is widely used as an indicator of environmental pollution: this social hymenopteran strongly interacts with vegetables, air, soil, and water surrounding the hive and, as a consequence, pollutants from these sources are translated to the insect and to the hive products. During the wide-ranging foraging activity, the forager bee is known to collect samples of the main airborne PM pollutants emitted from different sources and therefore it can be used as an efficient PM sampler. In the present research, PM contaminating forager bees living nearby a cement factory and several kilometers away from it has been analysed and characterised morphologically, dimensionally and chemically through SEM/EDX. This provided detailed information on the role of both the cement manufacturing activities and the vehicular traffic as sources of airborne PM. This may indeed help the implementation of appropriate preventive and corrective actions that would effectively minimize the environmental spread of pollutant PM not only in areas close to the plant, but also in more distant areas.

## INTRODUCTION

Industrial activities have a key role in the economic well-being of a country but they may involve processes with a more or less profound environmental impact, such as energy consumption, waste production and, above all, emission of pollutants into the atmosphere, water and land. Among industrial emissions, airborne particulate matter (PM) is of special concern, given that PM exposure is ubiquitous and linked with several adverse health outcomes (*Brauer et al., 2016*). PM is a complex mixture of airborne chemicals classified

Corresponding author
Ilaria Negri, ilaria.negri@unicatt.it

according to the diameter which may range from several micrometers (PM10) to a few nanometers (PM0.1). The dimension of PM plays a role in toxicity and pathogenesis, too: when inhaled, finer particles may penetrate deeper in the airways tract, cross the blood barrier and then distribute to most organs (*Nemmar, 2002*; *Valavanidis, Fiotakis & Vlachogianni, 2008*; *Ni, Chuang & Zuo, 2015*). Ultrafine particles may even enter the brain directly via the olfactory bulb, posing hazards to human health (*Maher et al., 2016*). The cement manufacturing industry is one of the main industrial sources of airborne particulate which can be discharged into the atmosphere at almost every stage of the process, from quarrying to handling, packing and transportation (*Abdul-Wahab, 2006*; *Rodrigues & Joekes, 2011*). To prepare cement, naturally occurring calcareous deposits, such as limestone, marl or chalk, are commonly used as sources of calcium carbonate, while quartz, clays and sand provide silica. As partial replacements for natural raw materials, other materials such as bauxite, iron ore and slag are often added in the blend, as well as several types of waste containing calcium, lime, aluminate, silicate and iron (*Rodrigues & Joekes, 2011*). After crushing, grinding and homogenization of the ingredients, the crude mixture is heated at very high temperature (about 1.450 °C) to form a granular material called clinker. Then clinker is milled together with gypsum and other constituents such as fly and pozzolanic ashes, and it undergoes a final grinding before packaging. Dust from cement manufacturing is made up of particulates of different sizes and a substantial part of it is made up of breathable PM. Cement dust may also contain pollutants, like heavy metals of toxicological relevance (*Schuhmacher, Domingo & Garreta, 2004*; *Schuhmacher, Nadal & Domingo, 2009*; *Rodrigues & Joekes, 2011*). As a consequence, cement dust exposure in workers with no adequate protective equipment or residents nearby the plant may cause severe adverse effects through skin or eye contact, inhalation and swallowing (*Spoo & Elsner, 2001*; *Manjula et al., 2013*; *Gizaw, Yifred & Tadesse, 2016*; *Egbe et al., 2016*). In this study, we analysed airborne PM collected by honey bees living in a rural hilly landscape (Val d'Arda, Piacenza, Italy), characterized by the presence of a cement factory as the main industrial settlement. This social hymenopteran is a well-known indicator of environmental pollution: the bee strongly interacts with vegetables, air, soil, and water surrounding the hive and, as a consequence, pollutants such as heavy metals, radionuclides and pesticides from these sources are translated to the insect and to the hive products (*Leita et al., 1996*; *Negri et al., 2015*; *Herrero-Latorre et al., 2017*; *Zarić et al., 2017*). In a previous work we demonstrated that forager bees are ideal tools for monitoring airborne PM (*Negri et al., 2015*). During the wide-ranging foraging activity, which takes place several hundred meters around the hive, the bees are able to collect samples of the main airborne particles emitted from different sources (*Negri et al., 2015*). Airborne dusts adhere to the insect's body (and specifically in peculiar regions of the fore-wings and the head) and can be characterized dimensionally, morphologically and chemically by means of a Scanning Electron Microscope (SEM) coupled with X-ray spectroscopy (EDX) (*Kutchko & Kim, 2006*; *Choël et al., 2007*; *Negri et al., 2015*). This may provide useful information on the different emission sources, both natural and anthropogenic. The aim of this study was to detect the presence of environmental markers linked to anthropogenic activities including

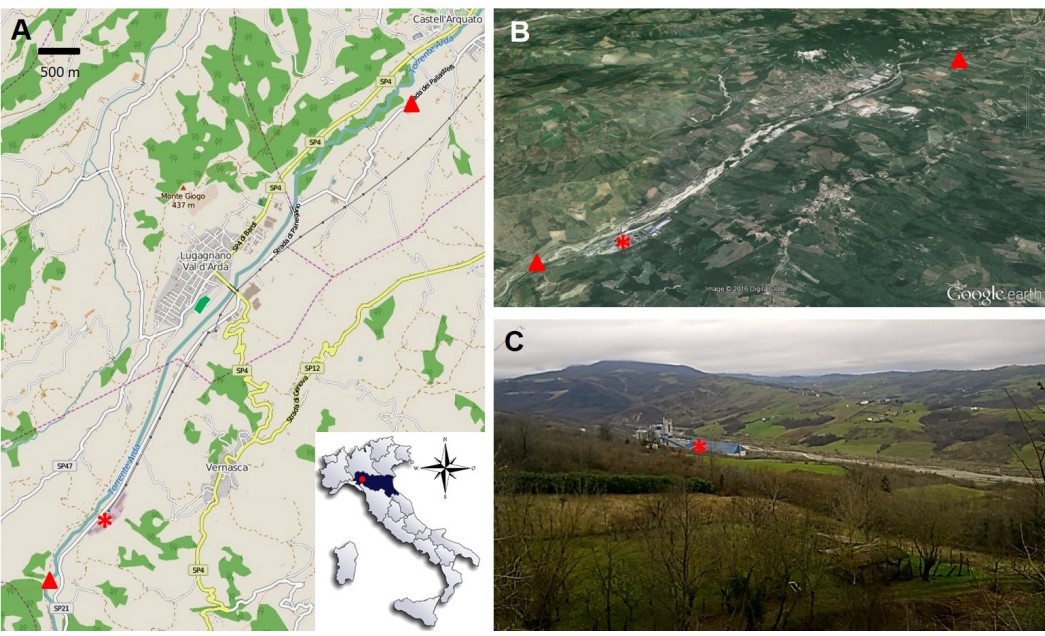

**Figure 1  Maps (A–B) and picture (C) of the area of investigation, showing the honey bee sampling sites (triangles) and the cement plant (asterisks).**

cement production and to highlight possible critical steps of the manufacture process responsible of dust escape.

## MATERIALS AND METHODS

### Area of investigation

The area of investigation is located in the Arda Valley (Fig. 1), a narrow valley in Piacenza province (North Italy), crossed by the Arda stream.

The valley is renowned for many naturalistic and touristic attractions, including the Mount Moria Park with centuries-old chestnut trees, the Natural Geological Reserve of Piacenziano rich in sedimentary rocks containing fossils, and the well-preserved medieval town of Castell'Arquato. About 15 Km from the source, the Arda stream is intersected by a dam, thus forming a basin which serves as water supply. A couple of kilometers away, just along the road which runs parallel to the stream, a cement factory was built in the late 1930's. The plant is currently 61.000 square meters in size and produces about 900.000 t/year of clinker (as a maximum nominal capacity) for the production of Portland cement. The cement plant is delimited by hills rich in badlands and partly exploited for farming of about 350 m in height from one side and 450 m from the other, and it lies about 1 Km from the little town of Vernasca, which is located on the main hills. Along the road through the valley lie the towns of Lugagano and Castell'Arquato, at about 3 and 7 Km far from the plant, respectively (Fig. 1).

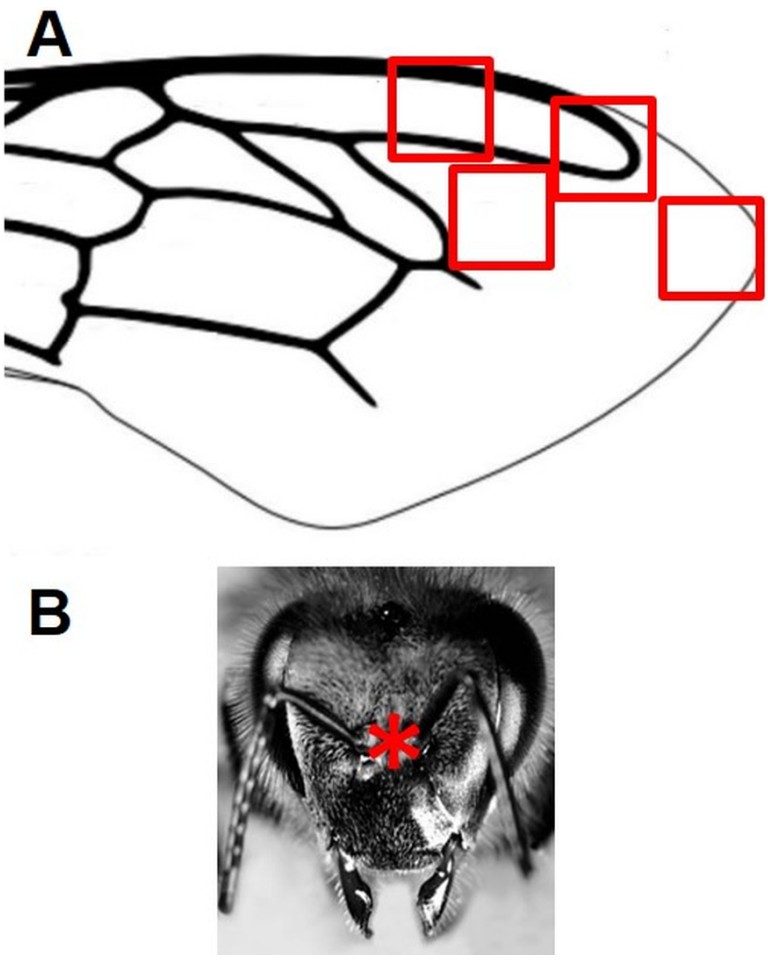

**Figure 2** Schematic representation of the areas of the wings (squares) (A) and the heads (asterisk) (B) of the bees subjected to analyses.

## Honey bee collection and analysis

At the end of April 2015, five worker bees (CP bees = Cement Plant bees) were sampled while foraging in a lawn along the provincial road n. 21, about 800 m south of the cement factory (Fig. 1). The apiaries were located nearby and all the bees came from the same colony. Bees collection was performed at noon during a sunny day, with a temperature of about 19 °C. Soon after, five foragers from a second colony were collected about 7 Km north from the cement plant, in a lawn close to the provincial road, between the town of Lugagnano and the little medieval village of Castell'Arquato (CA bees = Castell'Arquato bees; Fig. 1). The insects were prepared for SEM/EDX analyses as described in *Negri et al. (2015)*. After carbon coating, SEM-EDX measurements were carried out with a Tescan series VEGA. Secondary Electrons (SE) and Backscattered Electrons (BSE) images, as well as EDX point analyses, were acquired in alternating sequence at the same conditions of 20 kV with a nominal beam current of about 1 nA, in order to provide the chemical composition, morphology, surface characteristics and size of the particles. EDX spectra were interpreted

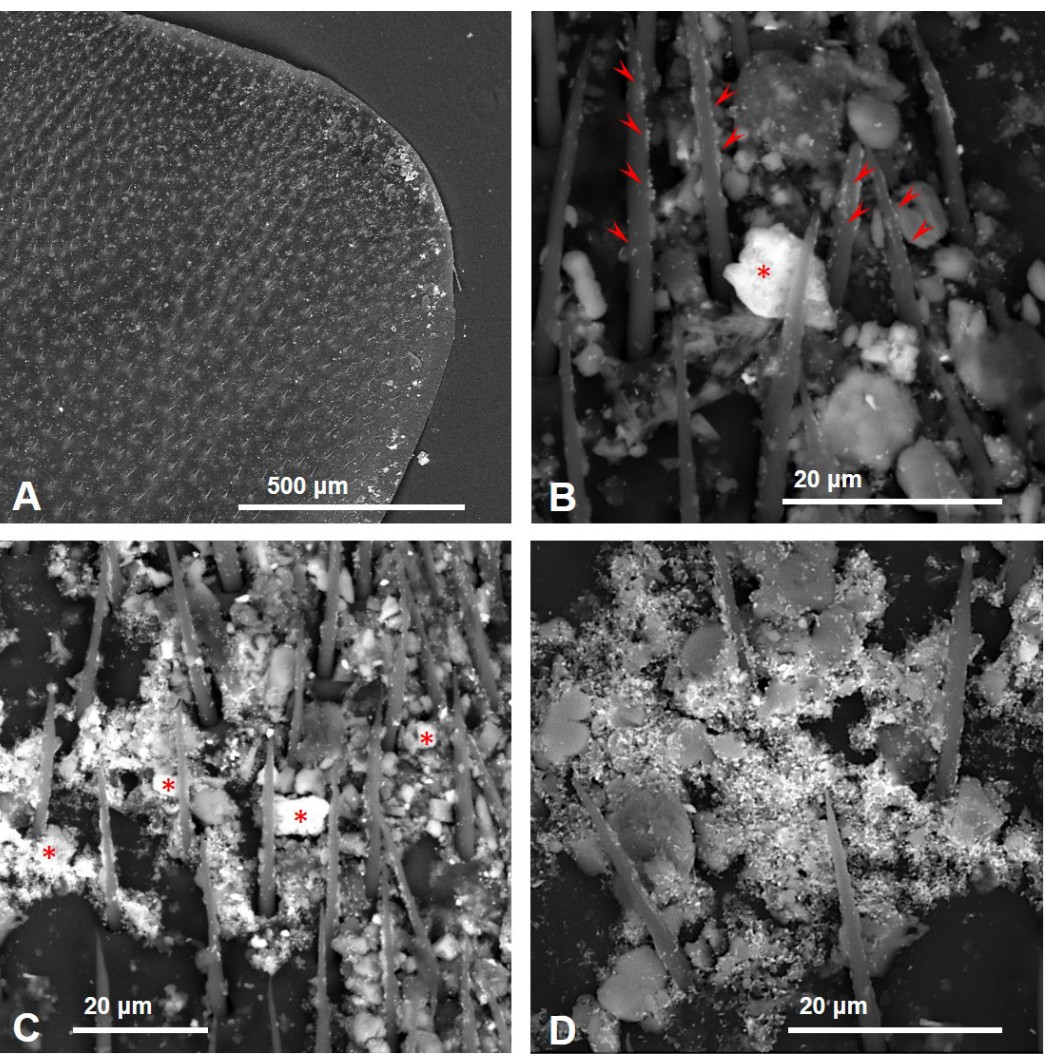

**Figure 3   SEM images of the fore wings of honey bees living close to the cement factory displaying PM (bright spots) concentrated along the costal margin of the wing (A), in some cases covering almost entirely the surface (B–D).** Asterisks, iron oxides/hydroxides; arrowheads, ultra-fine PM of baryte.

according to a mineralogical point of view, taking into account the mineral content of the surrounding geological formations. Where possible, single or multiple mineralogical phases have been identified. Measurements were carried out on particles present in 4 easily identifiable areas on the insect fore wings and 1 area of the head, 450 x 450 µm wide, as depicted in Fig. 2.

Then a rank for each area was assigned based on the level of contamination on a four-point scale, with ''1'' standing at the bottom of the scale and ''4'' at the top, namely 1, scarcely or not contaminated; 2, fairly contaminated; 3, highly contaminated; 4, heavily contaminated (Fig. S1). The non-parametric Mann–Whitney U test was then applied in order to statistically compare the contamination level of the bees collected nearby the cement plant (i.e., CP bees) and those collected about 7 Km away from it (i.e., CA bees).

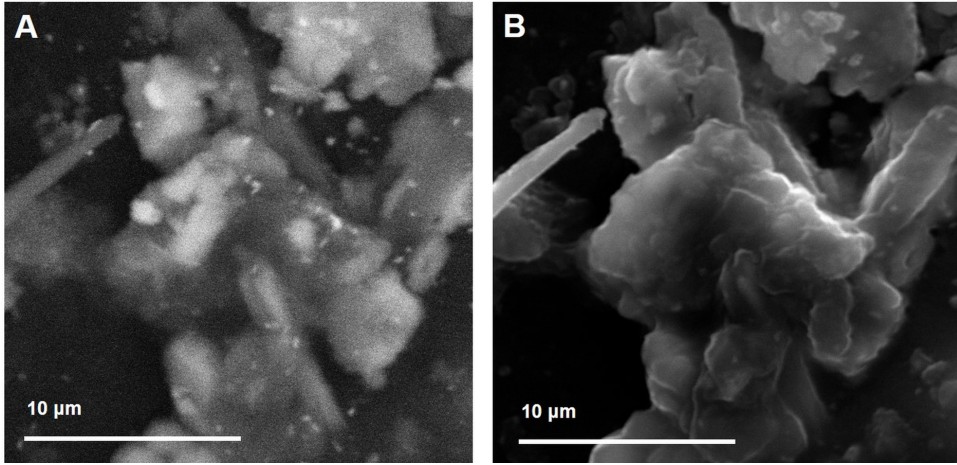

**Figure 4   Backscattered Electron (A) and Secondary Electron micrographs (B) of a complex aggregate of dusts.** While the BSE image provides details on the internal structure highlighting the presence of multiple chemical elements, the SE image reveals the complex topographic structure of the aggregate.

## RESULTS

Observations by means of Scanning Electron Microscope of bees collected close to the cement plant revealed the presence of numerous inorganic particles (Fig. 3). Some areas were almost entirely covered by the dust (Figs. 3B–3D) that sometimes appeared composed by complex aggregates, as shown in Fig. 4.

Inorganic particles were also found on the wings and heads of the insects collected about 7 Km far from the plant (Figs. 5 and 6). These bees appeared less contaminated than CP bees, at least for the areas of the wings (Mann–Whitney $U$ test $p \leq 0.01$; Fig. S2). Instead, the heads displayed a similar level of contamination in both CA and CP bees (Mann–Whitney U test $p = 0.07$; Fig. S2).

A list of chemical elements and corresponding mineralogical phases collected by CP and CA bees is shown in Table 1.

While the morphology of PM collected by CP bees was mostly irregular, some particles with symmetrical shapes have been found. Among them, round-shaped PM was occasionally observed (Fig. 7): few displayed a single hole (Fig. 7A) and the presence of Si, Al, K, Mg, O with traces of S and Fe; others were essentially made of iron oxides/hydroxides (Fig. 7B) or lead (Fig. 7C).

Halite (NaCl) crystals, displaying the typical cubic morphology (Fig. 8A), and angular-shaped fragments of silica glass were also present (Fig. 8B).

As far as the chemical composition of PM collected by CP bees, a large number of iron oxides/hydroxides of different size was present in all samples (Figs. 3B–3C). The composition of other dusts was compatible with that of mineralogical phases, such as gypsum ($CaSO_4$), calcite ($CaCO_3$), plagioclase ($NaAlSi_3O_8$–$CaAl_2Si_2O_8$), chlorite (($MgFeAl$)$8$($SiAl$)$8O20(OH)16$), quartz ($SiO_4$), fluorite ($CaF_2$), sylvite ($KCl$), portlandite ($Ca(OH)_2$), zircon ($ZrSiO_4$), xenotime ($YPO_4$), and baryte ($BaSO_4$) (Figs. 9 and 10).

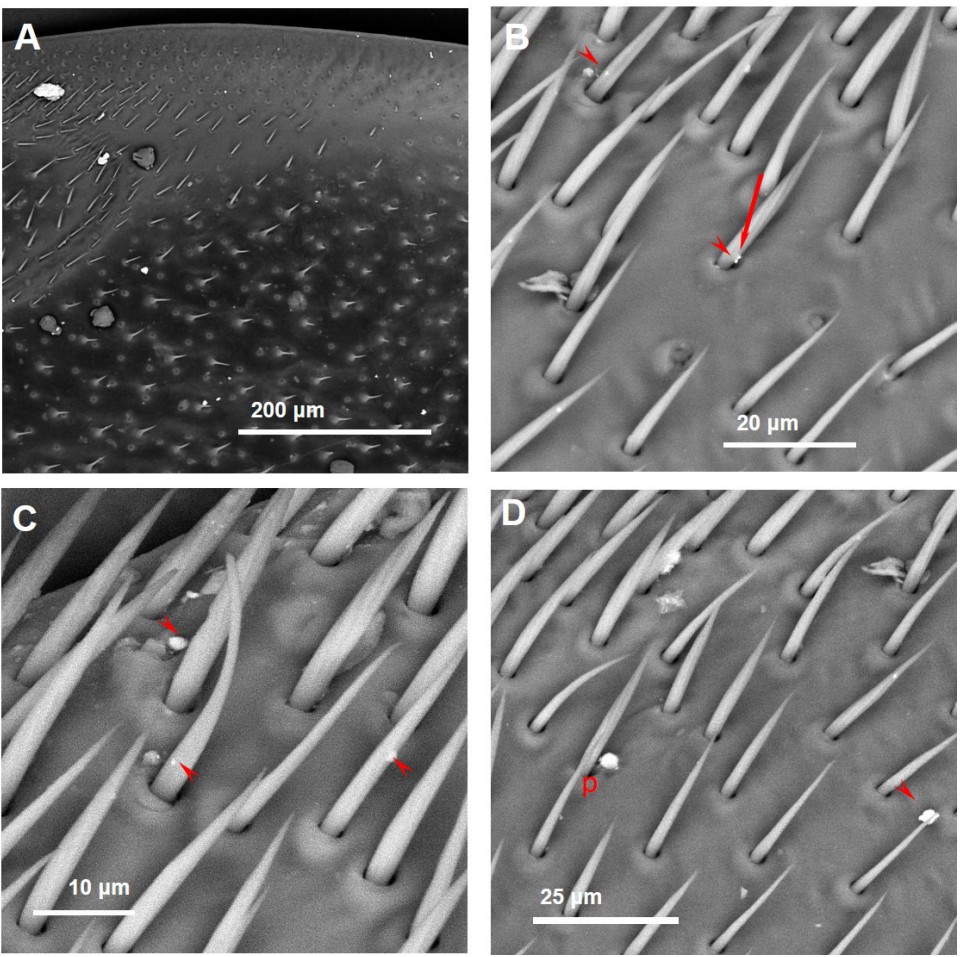

**Figure 5 Wings of honey bees collected 7 Km far from the cement plant.** (A) Pollen grains and inorganic particles (brighter spots). (B) Arrow = iron oxide/hydroxide on the wing setae of $0.5 \times 0.4\ \mu m$; arrowheads = baryte fragments. (C) Arrowheads = fly ashes. The dimensions of the two smaller ones are $1.3 \times 0.6\ \mu m$ and $0.6 \times 0.4\ \mu m$, respectively. (D) p, phyllosilicate; arrowhead, portlandite.

Baryte was present as ultrafine fragments scattered on the hairs and the surface of the bees or stuck to bigger particles (Figs. 3B–3D). CP foragers also collected many dusts with a chemical composition compatible with fly ashes (Fig. 11) and cement (Fig. 12).

Other dusts were composed by complex aggregates, therefore it was not possible to assign a definite chemical composition (Figs. 13–15). Heavy metals such as Pb, Cu, Ti, Bi, Cr, Mn, Sn, and Zn were concentrated in such aggregates, whose composition sometimes appeared different according to the point of analysis (Figs. 13–15).

CA bees displayed particles containing some clay minerals and quartz (Table 1). Moreover, iron oxides, calcite, baryte, fly ashes, portlandite, and cement were also present (Figs. 5 and 6; Table 1).

Finally, few calcium oxalate crystals ($CaC_2O_4$) were also present on both the head and the wings of the insects (Fig. 6D, Fig. S4).

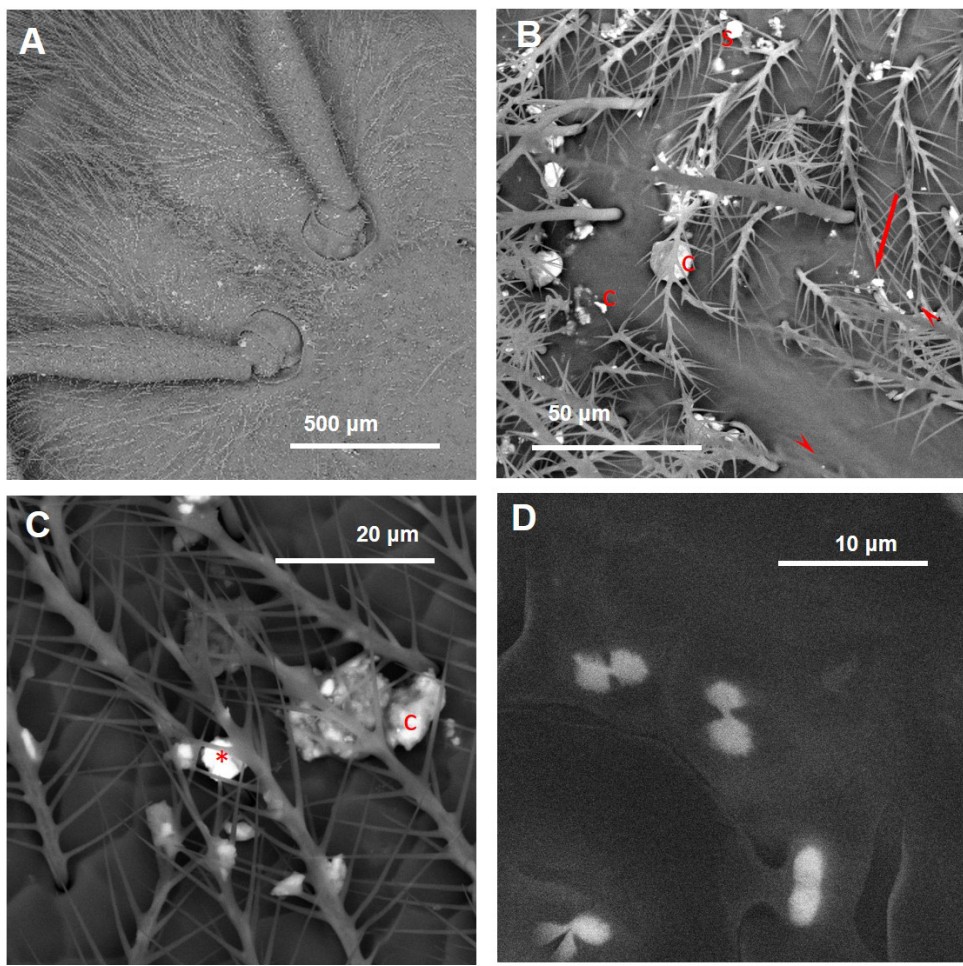

**Figure 6** **Inorganic particles carried by honey bees collected 7 Km far from the cement plant.** (A) Medial plane of the head displaying several inorganic dusts (bright spots). (B) c, cement; s, silica fume; arrowheads, phyllosilicate; arrow, quartz. (C) c, cement; asterisk: iron oxide/hydroxide. (D) Crystals of calcium oxalate on the bee wing.

## DISCUSSION

### Honey bee contamination

The worker bees living nearby the cement plant appeared significantly more contaminated by PM of natural and anthropogenic origin than bees sampled about 7 Km away from the plant. As far as minerals of natural origin, the bees all displayed the occasional presence of clay minerals and plagioclases, a group of common feldspar minerals, suggesting they should represent a negligible fraction in the composition of the local airborne PM. As far as anthropogenic dusts collected by both the bee populations, their chemical composition indicates that, besides vehicular traffic, the cement manufacturing activities represent the main source of airborne PM. This is not surprising since during the process of cement manufacture considerable amount of dust can be emitted at almost every stage and PM

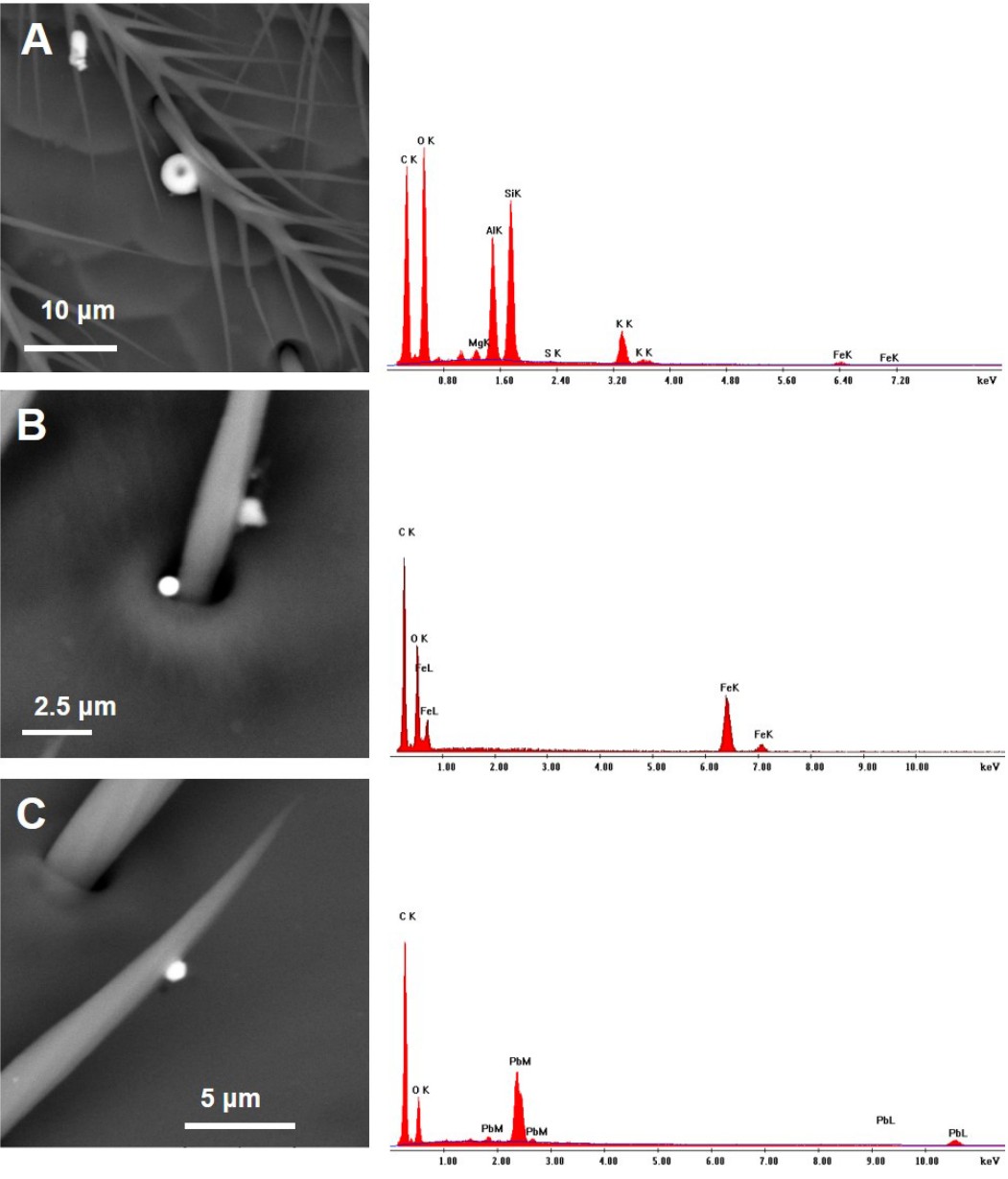

**Figure 7** **Round-shaped anthropogenic particles derived from high temperature combustions, stuck to honey bee setae or found inside hair sockets.** (A) Si-Al rich particle, 4,4 μm in diameter. (B) Particle of iron oxide/hydroxide of 0,7 μm. (C) Particle of lead of 0,9 μm.

can reach various distances mainly depending on topography, atmospheric conditions and particle size (*Abdul-Wahab, 2006*; *Branquinho et al., 2008*; *Schuhmacher, Nadal & Domingo, 2009*).

## PM from cement manufacturing activity

The chemical composition of several dusts carried by both CP and CA bees was compatible with cement and portlandite, i.e., calcium hydroxide which may derive from cement

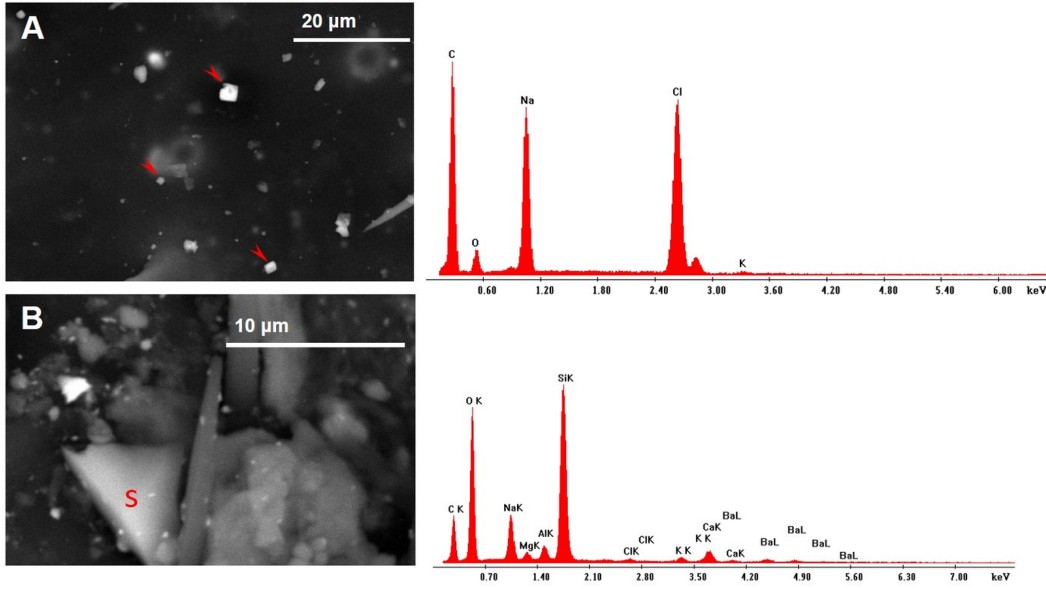

**Figure 8** Cubic crystals of NaCl (arrowheads; A) and a fragment of silica glass (s; B) found on the bee wings.

dust hydration when exposed to environmental humidity. The honey bees were also contaminated by many dusts whose composition is identical to that of the main ingredients for the cement blend. For instance, fine dusts of calcium carbonate were scattered on the wings and heads of both CA and CP insects. Although these minerals may be partly of natural origin, being present in the Arda Valley sediments and rocks (*Baldizzone, 2008*), they are used in great amount for cement preparation in form of fine dusts. All the bees carried fly ashes, which are added as source of silica in the blend, along with pozzolan (an amorphous fine grinded silica material) found on CP bees in form of silica glass particles. The use of sandstones and silica sand as raw materials may be responsible of the presence on the bees of quartz dusts, as well as zircon and xenotime. The origin of gypsum fragments analysed on CP bees may be both of natural and anthropogenic origin: it is indeed a natural constituent of clays, but in the plant CaSO4 is added to clinker in one of the final steps for cement preparation. The gypsum used in the factory may derive from flue gas desulfurization or even provided as a synthetic by-product of the reaction between the mineral fluorite with sulfuric acid. Interestingly, the presence on the honey bees of fluorite particles may account for the use in the factory of contaminated (synthetic) gypsum. A peculiar mineralogical phase collected by CP bees is halite. NaCl may indeed be present in wastes of alumina used as ingredients, since the production of aluminum and its alloys needs a saline mixture of NaCl and KCl for the protection and purification of aluminum from oxides and impurities during the melting process. Remarkably, some KCl (i.e., sylvite) fragments were collected by CP bees. Another possible origin of halite and sylvite salt emission from the plant is the chemical reactions occurring inside the cement kiln where chlorides react with alkalis present in raw materials and/or

**Table 1 Mineralogical phases and a list of elements found on dust collected by the honey bees.** In italics, the dusts also collected by bees living far from the cement plant (CA bees).

| Analyzed PM | Chemical elements[a] | | | | | | | | | | | | | | | |
|---|---|---|---|---|---|---|---|---|---|---|---|---|---|---|---|---|
| | Si | Al | Na | K | Fe | Ca | Mg | F | Ti | Ba | Pb | Zr | RE | P | Cl | S |
| Round-shaped[b] | Si | Al | – | K | Fe | – | Mg | – | – | – | – | – | – | – | – | S[d] |
| Round-shaped[b] | – | – | – | – | – | – | – | – | – | – | Pb | – | – | – | – | – |
| Round-shaped-iron ox/hydroxide[b] | – | – | – | – | Fe | – | – | – | – | – | – | – | – | – | – | – |
| Silica glass[b] | Si | – | – | – | – | – | – | – | – | – | – | – | – | – | Cl[d] | – |
| Sodium chloride[b] | – | – | Na | – | – | – | – | – | – | – | – | – | – | – | Cl | – |
| Iron ox/hydroxide[f,c] | – | – | – | – | Fe | – | – | – | – | – | – | – | – | – | – | – |
| Gypsum[f,c] | – | – | – | – | – | Ca | – | – | – | – | – | – | – | – | – | S |
| Sylvite[f,b] | – | – | – | K | – | – | – | – | – | – | – | – | – | – | Cl | S[d] |
| Calcite[f,c] | – | – | – | – | – | Ca | – | – | – | – | – | – | – | – | – | – |
| Plagioclase[f] | Si | Al | Na | – | – | Ca | – | – | – | – | – | – | – | – | – | – |
| Chlorite minerals[f] | Si | Al | – | – | Fe | – | Mg | – | – | – | – | – | – | – | – | – |
| Quartz[f,c] | Si | – | – | – | – | – | – | – | – | – | – | – | – | – | – | – |
| Fluorite[f,b] | – | – | – | – | – | Ca | – | F | – | – | – | – | – | – | – | – |
| Portlandite[b] | – | – | – | – | – | Ca | – | – | – | – | – | – | – | – | – | – |
| Zircon[b] | Si | – | – | – | – | – | – | – | – | – | – | Zr | – | – | – | – |
| Xenotime[b] | – | – | – | – | – | – | – | – | – | – | – | – | RE | P | – | – |
| Baryte[f,b] | – | – | – | – | – | – | – | – | – | Ba | – | – | – | – | – | S |
| Ash[b] | Si | Al | – | K | Fe | Ca | Mg | – | Ti[e] | – | – | – | – | P[d] | – | S[d] |
| Cement[b] | Si | Al | Na | K | Fe | Ca | Mg | – | Ti[e] | – | – | – | – | P[d] | – | S[d] |

**Notes.**

[a] C and O are not displayed as they are always present in EDX spectra, also because the dust leans on the organic matrix of the insect body.

[b] PM of anthropogenic origin.

[c] PM of possible natural and anthropogenic origin.

[d] Traces of P, Cl, S, possibly due to the presence of phosphates, chlorides, sulfates.

[e] Not present in some dusts.

[f] EDX spectra are provided as Figs. S3 and S4.

in the combustible, giving raise to gases of NaCl and KCl which leave the kiln and then condensate. Among other industrial dusts collected by CP bees are spherical particles (i.e., spheres of iron oxides/hydroxides, lead or mixtures of Si, Al, K, Mg, O, Fe and S), linked to high temperature combustions like those occurring in the cement kiln. However, we cannot exclude that spherical PM, such as coal combustion particles, are present among fly ashes used as additives to clinker. Therefore, they may represent fugitive emissions occurring during handling operations of cement ingredients or may be directly formed during the clinkering process that occurs at temperatures of about 1,450 °C, thus discharged to the atmosphere as channelled emissions.

## Multigrain aggregates

The presence of several dusts, mainly in forms of complex aggregates containing various heavy metals (e.g., Bi, Cr, Cu, Mn, Sn, Zn, and Pb) should be at least partly linked to the cement industry. Indeed, it is widely recognized that cement production is an important emission source of a wide range of heavy metals, most of which originates from raw materials (*Schuhmacher, Domingo & Garreta, 2004*; *Schuhmacher, Nadal & Domingo, 2009*;

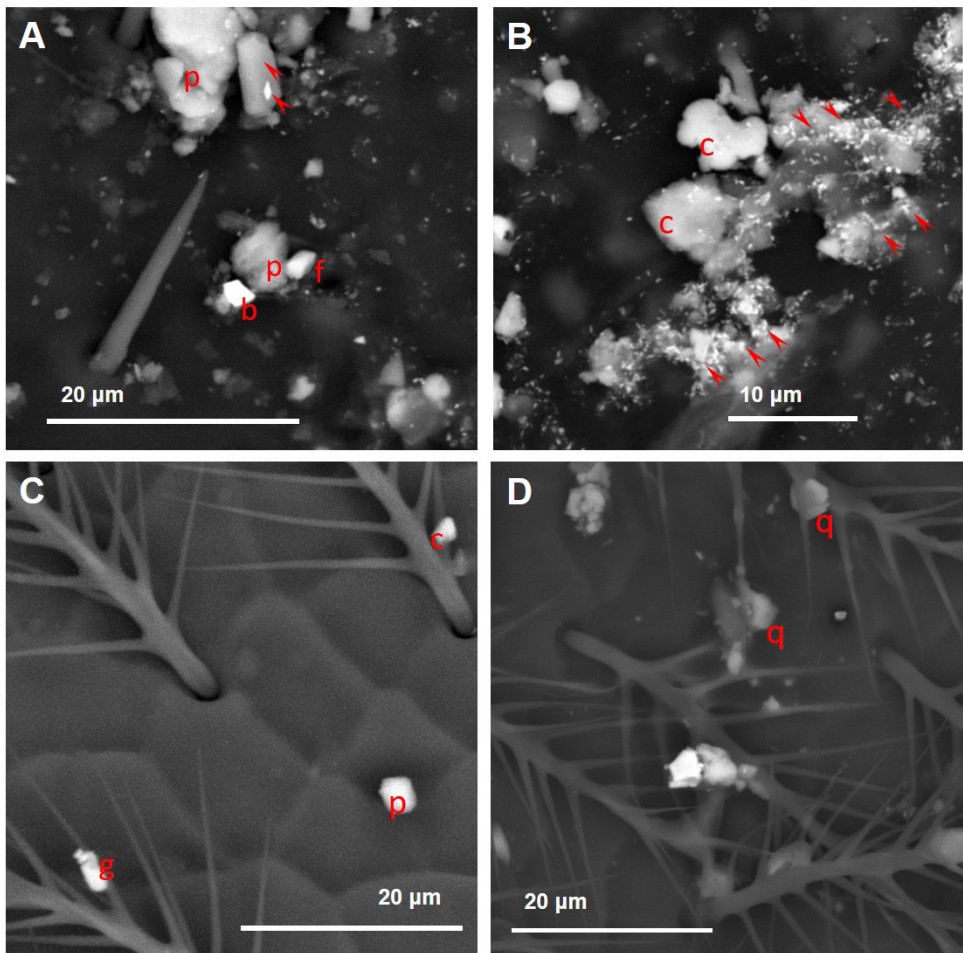

**Figure 9** **Variety of dusts observed on the bee wings (A, B) and heads (C, D) collected near the cement plant.** (A) arrowheads, plagioclase; p, phyllosilicate mineral; f, fluorite. (B) c, calcite; arrowheads, baryte. (C) g, gypsum; c, calcite; p, chlorite mineral. (D) q, quartz.

*Rodrigues & Joekes, 2011*), whereas for some heavy metals such as Cr, Cu, Zn, and Pb a contribution from tire and brake wear should also be considered (*Adamiec, Jarosz-Krzemińska & Wieszała, 2016*). As far as lead, it should be worth mentioning that this element is very persistent and its presence in airborne dust could also be a consequence of common use in the past of PbO4 as gasoline additive (*Adamiec, Jarosz-Krzemińska & Wieszała, 2016*).

## Origin of iron oxides/hydroxides

The presence of iron oxides/hydroxides dusts deserves special mention. Iron was observed covering large part of the body of all insects, and this contamination may originate both from the iron ore and slag (e.g., mill scale and pyrite clinker) used in the plant as additives for blending or also emitted from vehicle brakes (*Kukutschová et al., 2011*; *Grigoratos & Martini, 2015*). Indeed, this might also account for the presence of many iron oxides on CA bees, too, which live close to the main road of the valley, travelled not only by trucks

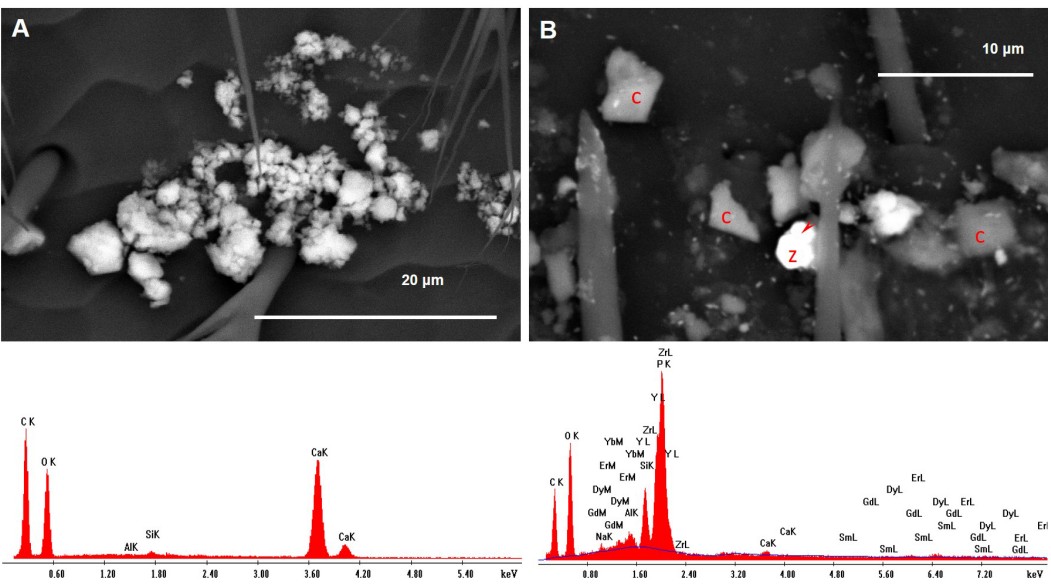

**Figure 10** **(A) Fragments of portlandite (calcium hydroxide); (B) fragment of zircon (z) with xenotime in epitaxy (arrowhead).** The area is also rich in fragments of calcite (c) and ultrafine PM of baryte (brighter spots).

carrying materials of the cement plant, but also by many other vehicles, including private cars, heavy and agricultural vehicles.

## Origin of barium sulfate

Another anthropogenic compound collected by both CP and CA bees is baryte, which spread on the bee wings in high amount and with numerous nano-sized fragments. This mineral is uncommon in the Arda Valley and mainly contained in rare septarian nodules (*Baldizzone, 2008*), then a natural origin of such a kind of dusts should be excluded. According to the literature, the occurrence of barium sulfate could be linked to vehicle traffic: indeed, $BaSO_4$ is used both in tires and brake pads for improving resistance, where the mineral accounts for nearly half percent of the total composition (*Österle et al., 2001*; *Adamiec, Jarosz-Krzemińska & Wieszała, 2016*). During braking simulation tests researchers demonstrated that baryte is released in form of many nano-sized fragments (*Österle et al., 2001*). It is worth mentioning that non-exhaust emissions usually represent one of the main sources of PM in urbanized areas where they account for over ninety percent of PM10 and eighty-five percent of PM2.5 emissions from traffic. Several studies also provide evidence that there is a positive correlation between vehicles' weight and non-exhaust emissions, even suggesting that the use of electric vehicles may not reduce levels of PM as much as expected, because of their relatively high weight (reviewed in *Timmers & Achten, 2016*). As a consequence, a significant role for heavy traffic in emitting non-exhaust PM in Arda Valley cannot be ruled out.
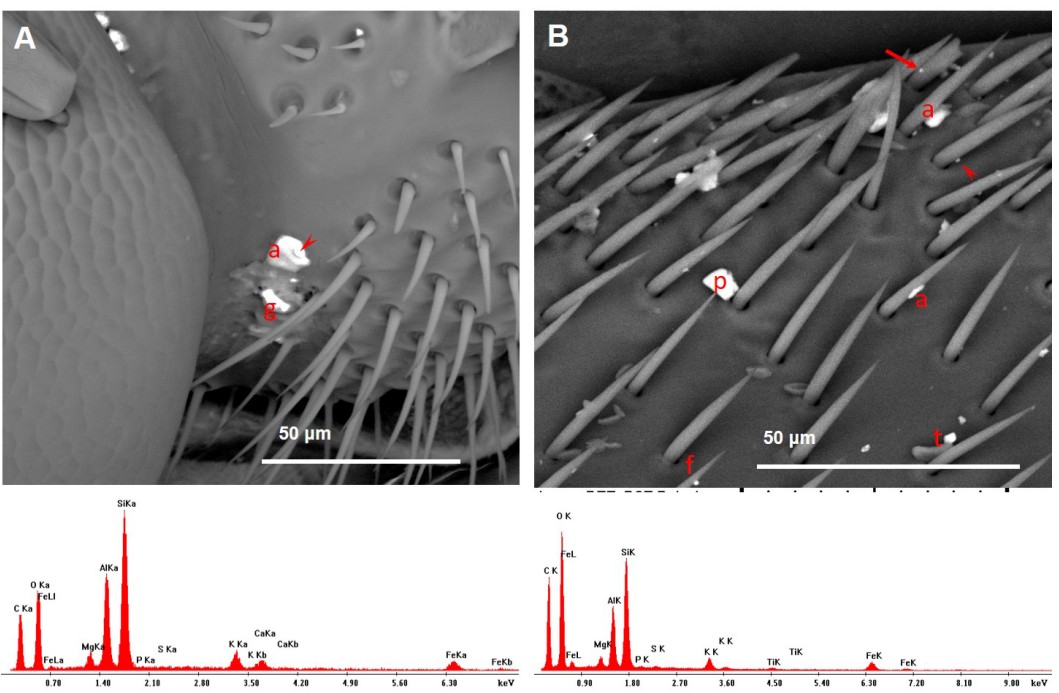

**Figure 11** **Variety of dusts observed on a bee head (A) and wings (B) collected near the cement plant.** Note the presence of fly ashes (EDX spectra). (A) An agglomerate of dusts on the antenna: a, fly ash; arrowhead, calcite; g, gypsum. (B) Many dusts were found stuck to the setae; a, fly ash; f, iron oxide/hydroxide; p, plagioclase; t, titanium oxide; arrow, small fragment containing lead and antimony; arrowhead, portlandite.

## PM from beekeeping activity

CA bees displayed the presence of calcium oxalate crystals which could derive from treatments carried by beekeepers on the bee family against the ectoparasitic mite *Varroa destructor*. Indeed, oxalic acid is commonly used as miticide, and the treatment is often carried out by dripping the solution from which crystals of oxalate may precipitate.

## Prevention and control of fugitive emissions

Even if cement manufacturing industry is one of the main industrial source of air pollutants, at present in the Arda Valley a permanent monitoring network is not available. With the exception of PM10, NO2 and O3, daily concentrations provided by a single ground-based monitoring device of the Regional Environmental Protection Agency (ARPAE), placed about 3,5 Km away from the cement factory in the town of Lugagnano (http://www.arpae.it/), indicated that no data are available on PM2.5 and finer particles discharged from the industrial activities as diffuse emissions. By using honey bees as monitoring tools for airborne PM we provided information on the environmental presence of dusts, even nano-sized, originated both from the cement manufacturing activities and the vehicular traffic. The considerable contamination level observed on CP bees raises some concerns on the application of the Directive 2010/75/EU of the European Parliament and the Council on industrial emissions (Integrated Pollution Prevention and Control) that

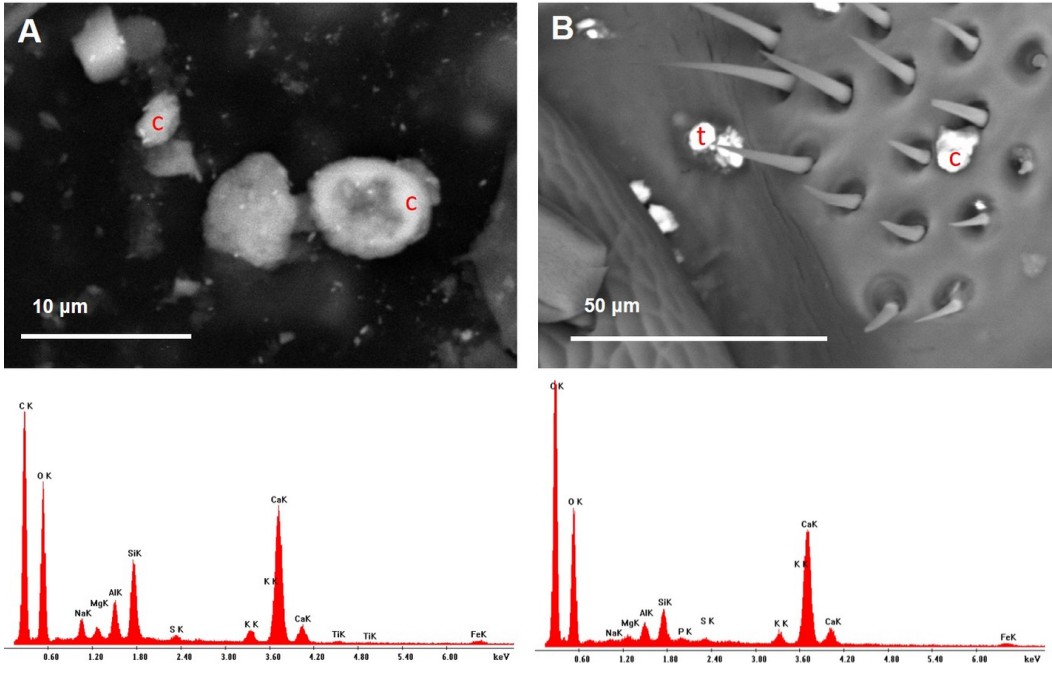

**Figure 12** **Variety of dusts observed on bee wings (A) and head (B) collected near the cement plant.**
Note the presence of cement dust (EDX spectra). (A) Bee wing with cement dusts. Ultrafine PM of baryte
(brighter spots) stuck to bigger dusts; c, cement. (B) Bee head with cement particle; c, cement; t, titanium
oxide.

lays down rules on integrated prevention and control of pollution arising from industrial
activities. In particular, according to the ''Best Available Techniques (BAT) Reference
Document for the Production of Cement, Lime and Magnesium Oxide'', the application of
a single BAT or a combination is enough to minimise/prevent diffuse dust emissions from
dusty operations and bulk storage. However, the contamination detected on the honey bees
suggests that even if many BAT have been set up in the cement plant, this does not appear
sufficient for the achievement of an effective prevention at least during the monitoring
period carried out in the present study. Even if a continuous monitoring of airborne PM
in the valley, covering all the productive season of the bees, could depict a more detailed
situation on the environmental contamination, our data suggest that handling of dusty
raw materials/final product may represent a critical step for dust leakage. Such operations
should always be followed by an accurate removal of any deposit on ground and facilities,
through wetting and removing mechanisms. The plant is indeed provided with sweeping
machines working for about 1000 hours/year (R Ing, F Buzzi-Unicem, pers. comm., 2017),
but cleaning should comprise not only the loading area inside the factory but also any
parking and manoeuvre areas used by trucks just outside the plant, which should be paved
for minimizing dust re-suspension. In addition to wind-blown, the spreading of dusty raw
materials/final product in the valley may also be explained by leakages from trucks getting
dirty after their activities inside the plant. Therefore, the implementation of a cleaning
system for trucks leaving the factory (e.g., a wheel washing system) should be set up. Finally,

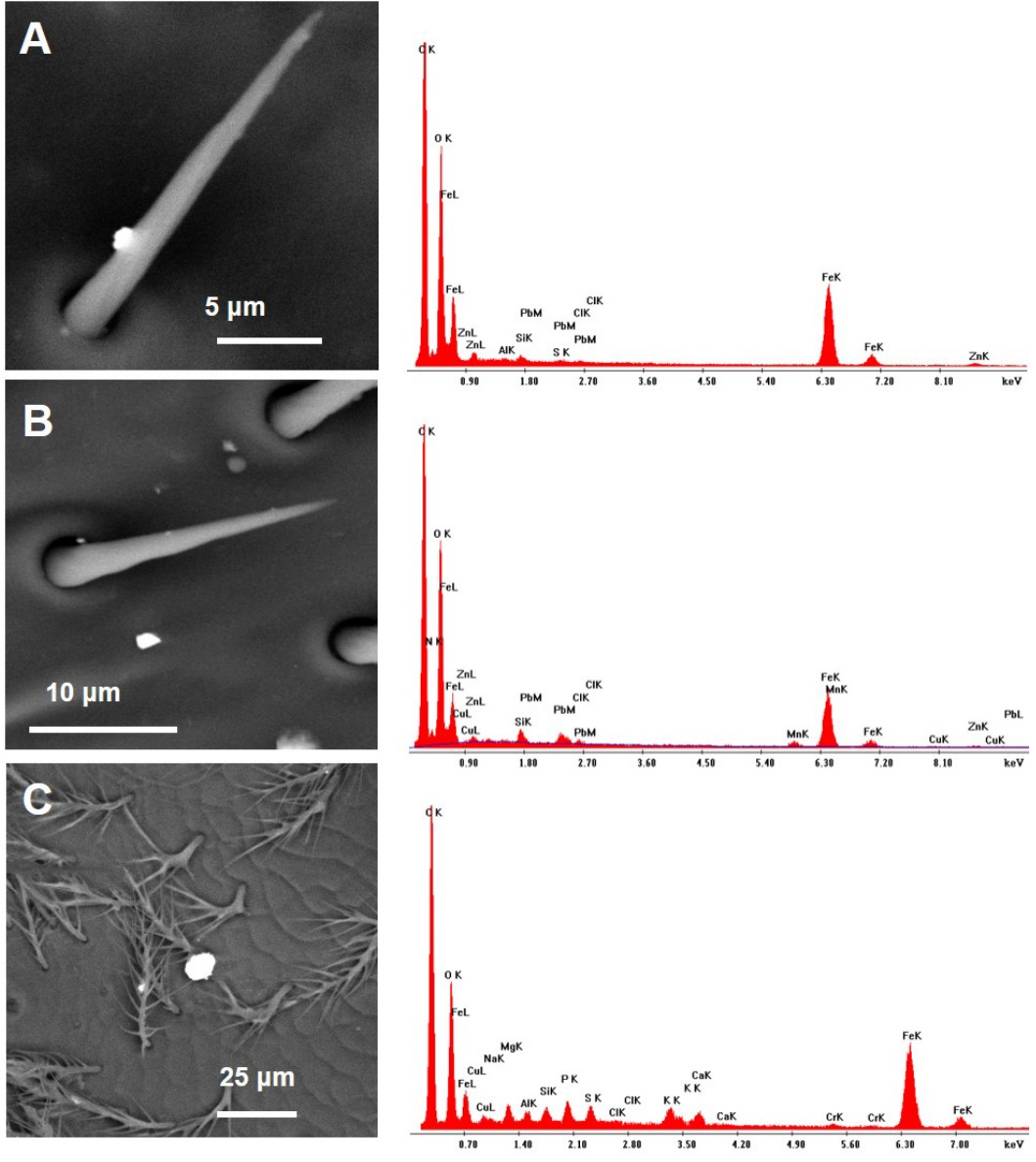

**Figure 13 Honey bees collected near the cement plant also carried several dusts with ambiguous chemical composition.** (A) PM1 attached to a seta of the bee wing. Note the presence in the EDX spectrum of lead, zinc and iron. (B) PM1.8 on the bee wing containing several elements, including copper, lead, zinc, iron and manganese. (C) PM6.5 on the bee head. Note the presence of many elements including chromium.

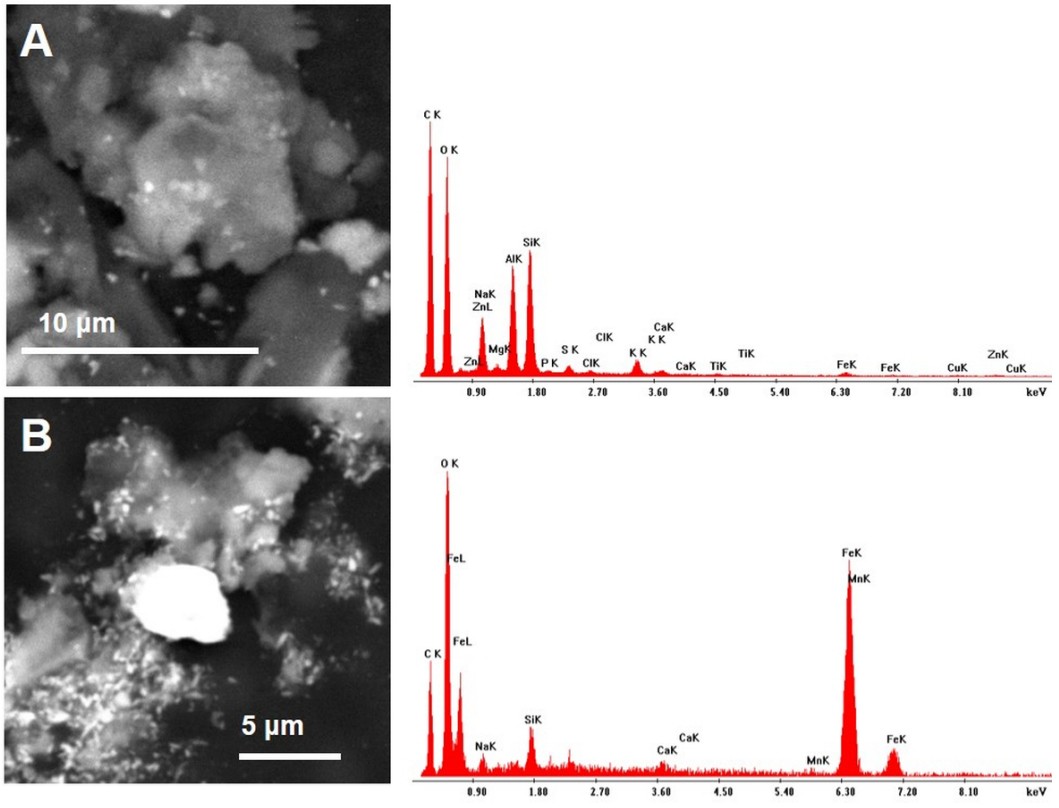

**Figure 14** **Aggregates of dusts observed on honey bees collected near the cement plant.** (A) An aggregate of dusts containing many elements, including iron, titanium, zinc and copper (EDX spectrum). Some ultra-fine PM of baryte are also visible (brighter spots). (B) A bright PM containing iron and manganese.

a regular sweeping followed by washing of the road run by the trucks may be a reliable practice to minimize the environmental impact of PM. This procedure might also help preventing dust re-suspensions from non-exhaust emissions from cement trucks, as well from other vehicles including private and agricultural machines.

## CONCLUSIONS

The use of drones for watching pollution levels supplementing the existing ground-based monitoring systems is growing. At this regard, the honey bee should be considered as a natural pollution-sensing drone able to provide the following advantages: (i) limited purchase costs and maintenance; (ii) a unique sampling system, (iii) an environmentally friendly approach; (iv) the simultaneous collection of a wide range of pollutants, including airborne particulate matter. In this study we give evidences that single particle analysis of airborne dusts collected by the bee provides accurate information on PM characteristics and on the specific contribution of different emission sources (e.g., industries and vehicular traffic). Replication of this study across seasons and other urban sites is readily applicable and may help the implementation of adequate control strategies in order to reduce the impact of pollutants on both the environment and public health.

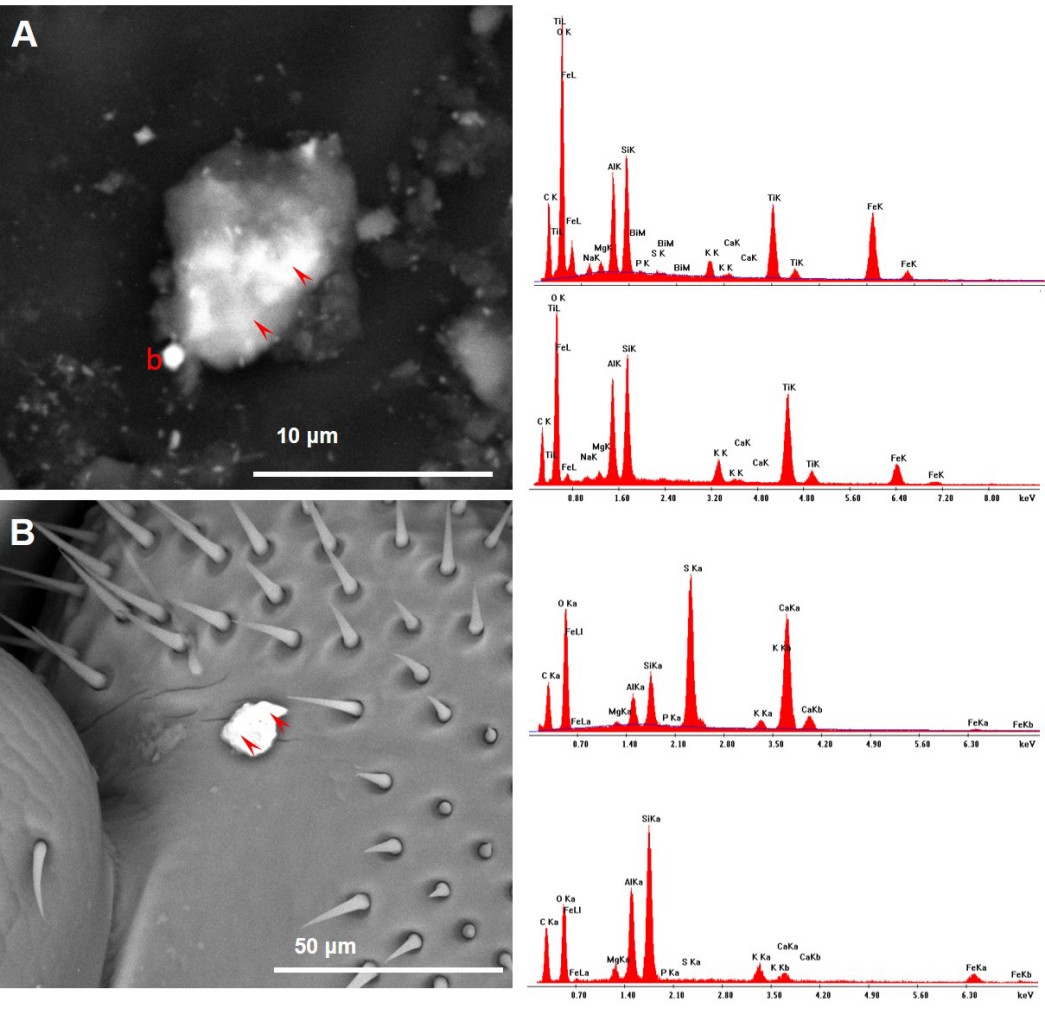

**Figure 15  Honey bees collected near the cement plant also displayed some dusts whose chemical composition appeared different according to the point of analysis.** (A) The arrowheads indicate the points of spectra acquisition (EDX spectra on the right): the brighter part of the aggregate showed the presence of bismuth, that was lacking in the darker one; b = PM0.8 of baryte. (B) The arrowheads indicate the points of spectra acquisition (EDX spectra on the right): the upper part of the dust was mainly composed by gypsum; the lower part has a composition compatible with a fly ash.

## ACKNOWLEDGEMENTS

We would like to thank Dr. Paolo Gentile (Università Milano Bicocca, Italy) for SEM/EDX analysis and help in EDX spectra interpretation.

### Funding

The study was funded by Koine'—Consulenze ambientali Snc, Montechiarugolo (Parma), Italy. The funders had no role in study design, data collection and analysis, decision to publish, or preparation of the manuscript.

### Grant Disclosures

The following grant information was disclosed by the authors:
Koine'—Consulenze ambientali Snc, Montechiarugolo (Parma), Italy.

### Competing Interests

Ilaria Negri is an Academic Editor for PeerJ. Marco Pellecchia is an employee of Koiné—Environmental Consultancy, Montechiarugolo (Parma), Italy.

### Author Contributions

- Marco Pellecchia conceived and designed the experiments, performed the experiments, analyzed the data, contributed reagents/materials/analysis tools, authored or reviewed drafts of the paper, approved the final draft.
- Ilaria Negri conceived and designed the experiments, performed the experiments, analyzed the data, prepared figures and/or tables, authored or reviewed drafts of the paper, approved the final draft.

### Data Availability

 The raw data are provided as Fig. S1–Fig. S4.

### Supplemental Information

Supplemental information for this article can be found online at http://dx.doi.org/10.7717/peerj.5322#supplemental-information.

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
