# Peer review of "Particulate matter collection by honey bees (Apis mellifera, L.) near to a cement factory in Italy"

_PeerJ, doi:10.7717/peerj.5322_

## Round 0.1 · original submission · Major Revisions

The manuscript establishes a very promising approach to the use of bees as samplers of particulate matter in the environment. However, the lack of replication is insufficient to test the validity of your method. Additional data are required for a favorable decision to publish.

Reviewer 1 ·

Basic reporting

Language needs to be thoroughly revised. I could follow the text but with great difficulty.

The introduction section nicely explains PM, its origin in the cement factory and its possible adverse health effects. However, considering that in this paper honeybees are used as samplers of PM it would be good to have a few sentences about the use of honeybees as bioindicators.

Some things in the text should be better defined. For example:
“Several studies also provide evidence that there is a positive correlation between weight and non- exhaust emissions, even suggesting that the use of electric vehicles may emit comparable level of PM as conventional vehicles because of their relatively high weight (reviewed in Timmers and Achten, 2016).”
Here it should be pointed out that weight refers to the weight of the car.

In the section 3.2 Environmental impact of PM the authors say little that is in accordance with the title of this section. Instead this section mainly has suggestions to the plant management on how to reduce the spreading of PM. Although this is very useful and should remain a part of this paper it should not be in the section with the above title.

Experimental design

My biggest concern about the design of the experiment is that the bees were sampled only once. It would have been more useful to know if there were any changes in PM collected by bees depending on different months or seasons. However, the research done proves that honeybees in combination with the method applied can be used for long term monitoring of the pollution in the region. This greatly improves the knowledge of the scientific community on the use of honeybees as samplers of PM.

Specific comments:

“Measurements were carried out on particles present in distinctive areas on the insect wings and head, chosen as standard areas. Specifically, 4 areas of the wings and 1 area of the head wide 450 x 450 µm have been considered (Figure 2).”
It would be useful to include an explanation why where those 4 areas of the wing chosen as standard areas.

Validity of the findings

The conclusions are too general. The authors should focus in short on the findings presented in the paper, and then go on to general remarks.

Reviewer 2 ·

Basic reporting

The manuscript is presented in reasonably good scientific English. However, there are cases where word choice seems strange to me. I have attempted to highlight problematic areas and make language suggestions in the attached pdf.

The introduction provides sufficient background on the use of honey bees as environmental samplers as well as some background on the process of making cement. The intro is informative without being overly long.

There are too many figures and, while reference in the manuscript is made to “supplemental materials”, it is not clear which of the figures are destined for “Supplemental”. The figures themselves, however, are high quality and add to the manuscript -- there are just too many of them. I would really have liked to have seen example images that could help me understand what the 4 different levels of particulate contamination look like.

Experimental design

The research question, whether bees can be used as bioindicators for particulate matter, is an interesting and important question that the manuscript does a fair job of beginning to address.

The most serious criticism I have of this paper is that results came from 10 bees collected from 2 sites (5 per site) at a single time point. While the work appears to be performed to a high technical standard with the use of SEM/EDX, this is simply not enough replication to allow for generalizable conclusions to be drawn. As a proof-of-concept paper I think this would be valid, but more replication across time is needed to demonstrate that the observed patterns hold up to replication.

The methods are, for the most part, well described and should make it possible for the study to be replicated. I was a bit confused about which parts of the bee were studied from the description in the text (which should be reworded), but Fig. 2 makes it clear what was being studied.

Validity of the findings

Given the low level of replication, the only statistics it is possible to do were the comparison of contamination levels between bees and between sites. It could be argued that using individual bees in the site comparison is pseudoreplication and multiple samplings at different time points should really be used.

The discussion is overly long and should be reduced. Given that the statistical inference is preliminary, given the low level of replication, the discussion delves into the more anecdotal nature of the results. While some discussion of this nature is welcome, particularly as it relates to possible emissions from the cement plant and roadways, I found it to be overly long.

Annotated reviews are not available for download in order to protect the identity of reviewers who chose to remain anonymous.

---

## Round 0.2 · Minor Revisions

The primary objection of reviewer two that replication is lacking (n = 10, 2 sites, single time point) has been addressed along with the other points of both reviewers. I infer from revisions that no additional bee sampling is possible for this manuscript. Despite lack of replication this work is well done and should be published by PeerJ.

Emphasis on the uniqueness of the bee sampling system relative to other stationary methods should be emphasized in the discussion. For example, are bees integrated collectors for increased variety and chemical species of PM? Are bees more efficient and effective than other PM samplers. Is cost a factor--bees are a product of nature and essentially free. The case that bees are of high value for PM collection is well made and should be expanded with follow on work. Replication across seasons and other sites by diverse researchers across urban areas is readily applicable.

---

## Round 0.3 · accepted · Accept

Thank you for addressing the points raised by the reviewers throughout the review process. I have closely followed the tracked changes and find your revisions to be in order. Your manuscript is accepted for publication in PeerJ.

I note the following improvements and comments:

1. Language has been revised for improved readability. Word choice has been revised according to reviewer suggestions noted in the pdf of reviewer 2.

2. The introduction has been revised to include text regarding the use of honeybees as bioindicators.

3. Some things in the text should be better defined. For example:
“Several studies also provide evidence that there is a positive correlation between weight and non- exhaust emissions, even suggesting that the use of electric vehicles may emit comparable level of PM as conventional vehicles because of their relatively high weight (reviewed in Timmers and Achten, 2016).”
Here it should be pointed out that weight refers to the weight of the car. The clarification has been made in the revised text.

4. In the section 3.2 Environmental impact of PM the authors say little that is in accordance with the title of this section. Instead this section mainly has suggestions to the plant management on how to reduce the spreading of PM. Although this is very useful and should remain a part of this paper it should not be in the section with the above title. This section has been amended and the overall length shortened considerably.

5. The manuscript has been re-worded, specifically in the conclusion section, to emphasize that replication represents a gap in the study and is not the focus of the study. However, the research done proves that honeybees in combination with the method applied can be used for long term monitoring of the pollution in the region. This greatly improves the knowledge of the scientific community on the use of honeybees as samplers of PM and highlights the improvements needed to demonstrate replication of the initial findings.

6. “Measurements were carried out on particles present in distinctive areas on the insect wings and head, chosen as standard areas. Specifically, 4 areas of the wings and 1 area of the head wide 450 x 450 µm have been considered (Figure 2).” It would be useful to include an explanation why where those 4 areas of the wing chosen as standard areas. An explanation has been added addressing this point and a comparison of the levels of contamination has been include in supplementary figure 1.

7. The number of figures has been reduced from 16 to 15. Supplemental files consisting of four figures are informative. All of the figures of are of high quality and provide visual information on the diverse contamination found in the study. Whereas replication lacks, the strength of this study is in detailed illustration of in situ contaminant particles on the bee anatomy clearly establishing the approach as a unique environmental biosensor.

#